# Belt-type electrical muscle stimulation preserves muscle fiber size but does not improve muscle function in a rat model of cancer cachexia

**Karina Kouzaki**[1]*, **Mako Isemura**[2], **Yuki Tamura**[1], **Hiroyuki Uno**[1,2], **Shunta Tadano**[2], **Ryuji Akimoto**[2], **Katsu Hosoki**[2], **Koichi Nakazato**[1]

**1** Graduate School of Health and Sport Science, Nippon Sport Science University, Setagaya-ku, Tokyo, Japan, **2** HOMER ION Laboratory Co., Ltd., Shibuya-ku, Tokyo, Japan

* kouzaki@nittai.ac.jp

## Abstract

Cancer cachexia causes severe muscle wasting, and current treatments remain limited. Belt-type electrical muscle stimulation (bEMS) has emerged as a passive rehabilitation tool capable of activating multiple lower limb muscles simultaneously. We investigated whether bEMS prevents muscle wasting and improves functional outcomes in rats with cancer cachexia. Cancer cachexia was induced in male Sprague-Dawley rats by intraperitoneal injection of AH130 Yoshida hepatoma cells. Acute and chronic effects of bEMS were tested. Muscle protein synthesis was evaluated using the SUnSET method, and muscle fiber cross-sectional area (CSA) and ankle torque were measured after chronic stimulation. bEMS increased puromycin-labeled protein levels on day 3 post-injection (~1.5–2.0 fold; $p < 0.05$). After 10 days, bEMS mitigated reductions in muscle CSA in the gastrocnemius and tibialis anterior compared to the cachexia group. However, muscle strength (ankle torque) was not significantly improved. bEMS preserved muscle fiber size in cancer cachexia but failed to restore muscle function. These findings suggest bEMS may serve as a supportive strategy against muscle atrophy in cachectic conditions.

## Introduction

Cachexia is a multifactorial syndrome frequently observed in patients with advanced cancer, heart failure, chronic kidney disease, and sepsis [1–4]. It is characterized by unintended weight loss, systemic inflammation, anorexia [5], and significant reductions in both adipose and skeletal muscle mass [6,7]. Among these manifestations, skeletal muscle wasting is a critical concern due to its direct association with functional decline, increased morbidity, and reduced survival. In cancer patients, cachexia affects approximately 50–80% of individuals [8], compared to 5–15% in other chronic diseases [9]. The syndrome is driven by an imbalance between protein synthesis

**Data availability statement:** All relevant data are within the manuscript and its Supporting Information files.

**Funding:** This study was supported by a research grant from HOMER ION Laboratory Co., Ltd. (to KN) and a Grant-in-Aid for Scientific Research (B) from the Japan Society for the Promotion of Science (JSPS KAKENHI Grant Number 20H04041) (to KN). The funders had a role in study design, data collection and analysis, but the decision to publish and preparation of the manuscript were made independently of the funder, beyond the contributions of the authors employed by the funder. Specifically, the employees of HOMER ION Co., Ltd. contributed to the study as described in the 'Author Contributions' section.

**Competing interests:** I have read the journal's policy and the authors of this manuscript have the following competing interests: M. Isemura, H. Uno, S. Tadano, R. Akimoto, and K. Hosoki are employees of HOMER ION Co., Ltd. This does not alter our adherence to PLOS ONE policies on sharing data and materials. The other authors declare that they have no competing interests.

and degradation, with increased proteolysis [10,11] and impaired anabolic signaling contributing to progressive muscle atrophy [12].

Electrical muscle stimulation (EMS) is a non-pharmacological intervention capable of eliciting muscle contractions in the absence of voluntary effort. This feature makes it particularly useful for patients who are unable or unwilling to engage in physical exercise. Belt-type electrical muscle stimulation (bEMS) is a technique that enables the simultaneous activation of multiple lower limb muscle groups [13] and has been explored in rehabilitation settings [14,15]. Despite promising results in other populations, the effectiveness of bEMS in attenuating muscle wasting in cancer cachexia remains poorly understood. Previous studies in animal models have shown that EMS can enhance protein synthesis and promote muscle hypertrophy [16–19]. However, the clinical applicability of these findings remains uncertain, especially under cachectic conditions.

The present study aimed to investigate the effects of bEMS on skeletal muscle atrophy in a rat model of cancer cachexia. The AH130 Yoshida ascites hepatoma cell line is a well-established model known to induce rapid and severe cachexia, characterized by profound muscle and adipose tissue wasting in rats [20]. We evaluated the impact of acute and chronic bEMS on protein synthesis, muscle fiber cross-sectional area (CSA), and muscle function in the gastrocnemius and tibialis anterior muscles. The acute protocol was designed to determine whether a single session of bEMS could elicit an immediate anabolic response (i.e., muscle protein synthesis) to provide mechanistic insight. The chronic protocol aimed to evaluate whether repeated bEMS sessions over 10 days could translate into long-term structural (muscle fiber CSA) and functional (muscle torque) adaptations. These findings may provide insight into the utility of bEMS as a supportive intervention in cachexia management.

## Materials and methods

### Ethical approval and animal welfare

All procedures were approved by the Animal Experimental Committee of Nippon Sport Science University (Approval No. 017-A04) and conducted in accordance with the Fundamental Guidelines for Proper Conduct of Animal Experiments in Academic Research Institutions (MEXT, Japan; 2006, No. 71). The study adhered to ARRIVE guidelines. The study involved both acute and chronic experimental protocols. A total of 33 rats were used. For the acute study, animals were euthanized on day 3 or 5 post-injection (n = 5 per timepoint). For the chronic study, 23 rats were used (Saline: n = 7, AH130: n = 8, AH130 + Stim: n = 8) over a period of 10 days.

Throughout all experiments, animal health and general behavior were monitored daily, and body weight was recorded every other day. To minimize suffering, animals were housed in a controlled environment and all invasive procedures were performed under isoflurane anesthesia. The following pre-defined criteria were used as humane endpoints: (1) loss of more than 25% of the initial body weight; (2) development of severe ascites that visibly impaired mobility; or (3) a moribund state. During the chronic study, 2 animals in the AH130 group were euthanized prior to the study endpoint upon meeting humane endpoint criteria. No animals were found dead

unexpectedly in either study. At the end of each experiment, or when humane endpoints were met, animals were euthanized under deep isoflurane anesthesia (2.5% or higher) followed by exsanguination and tissue collection. All research staff involved in animal procedures have been trained and certified in proper animal handling and euthanasia techniques.

## Animals

Ten-week-old male Sprague-Dawley rats were obtained from CLEA Japan (Tokyo) and SLC Japan (Shizuoka). Rats were housed individually in a temperature-controlled room (23°C) with a 12 h light/dark cycle and provided ad libitum access to standard chow (CE-7; CLEA Japan) and water. Animals were acclimated for one week prior to experiments.

## Induction of cancer cachexia

Cancer cachexia was induced by intraperitoneal injection of AH130 Yoshida ascites hepatoma cells ($1 \times 10^8$ cells) under isoflurane anesthesia (2.0–2.2%; 250–300 mL/min) [21]. Cells were obtained from Tohoku University (ID: 0530, Cell Source Center for Biomedical Research · Cell Bank, Tohoku University, Sendai, Miyagi, Japan). Control animals received sterile saline.

## Electrical stimulation (bEMS)

The stimulation method was based on previous studies in animals and humans [13,19]. Hair was removed from the hindlimbs under isoflurane anesthesia. Rats were positioned supine and a bEMS device (Hormer Ion Corp., Tokyo, Japan) was attached to the knee and ankle joints. The knee and ankle joints were placed in a neutral position (Fig 1). They were not immobilized, allowing for joint movement (i.e., plantarflexion and dorsiflexion). Therefore, the electrically induced contractions

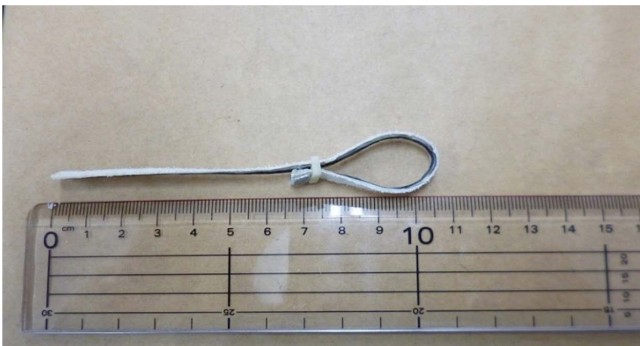

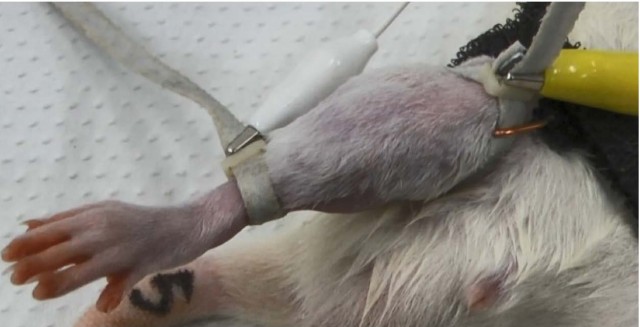

**Fig 1. Application of the belt-type electrode system.** Anesthetized rats were placed in a supine position after hair was removed from the lower extremities. Belt electrodes for electrical muscle stimulation were attached over the ankle and knee joints of the hind limbs.

were primarily isotonic. Electrical stimulation was applied at 60 Hz for 5 minutes to induce continuous tetanic contractions. A key characteristic of this bEMS setup is the simultaneous stimulation of antagonist muscles (e.g., tibialis anterior) along with agonist muscles (e.g., gastrocnemius), which may result in co-contraction and theoretically reduce the net joint torque.

## Experimental design

Acute response protocol: bEMS was applied to the right lower limb at either 3 or 5 days after AH130 injection (n = 5 per group). The left limb served as an internal control. MG and TA muscles were harvested 6 h after stimulation. In the acute response protocol, bEMS was applied unilaterally, allowing the contralateral limb to serve as an internal control for assessing immediate molecular responses. Chronic response protocol: Rats were divided into saline (n = 7), AH130 (non-stimulated, n = 8), and AH130 + Stim (bEMS-treated, n = 8) groups. bEMS was applied bilaterally every other day for 10 days starting one day after tumor cell injection. Muscles were harvested 48 h after the final stimulation. In the chronic response protocol, bEMS was applied bilaterally to evaluate long-term adaptive outcomes and to avoid potential confounding effects of a long-term unilateral intervention (Fig 2).

## Tissue collection and preparation

The medial gastrocnemius (MG) and tibialis anterior (TA) muscles were selected for analysis as they represent a key functional antagonist pair at the ankle and are well-suited for reliable dissection and tissue sampling. MG and TA muscles were excised and divided for histological and biochemical analysis. For CSA measurements, muscles were embedded in O.C.T. compound and frozen in cooled isopentane. Remaining tissue was frozen in liquid nitrogen for western blotting. Inguinal white adipose tissue (iWAT) was also collected to assess cachexia status.

## Western blotting

Muscles were homogenized in RIPA buffer with protease and phosphatase inhibitors. Protein concentration was measured via BCA assay. Samples (10 µg) were subjected to SDS-PAGE (10–12% gels) and transferred to PVDF membranes. After blocking, membranes were incubated with primary antibodies against phospho-p70S6K (Thr389; #9205), p70S6K (#9202), phospho-rpS6 (Ser240/244; #2215), rpS6 (#2217, Cell Signaling Technology), and puromycin (MAB343, Millipore), followed by Anti-rabbit IgG (#7074) or anti-mouse IgG (#7076, Cell Signaling Technology), HRP-linked secondary antibodies. Signals were visualized with chemiluminescence and quantified using ChemiDoc XRS and Quantity One software. Ponceau S staining was used as the loading control [22].

## Assessment of protein synthesis (SUnSET)

Puromycin (0.04 µmol/g body weight) was administered intraperitoneally 15 min before muscle collection [23,24]. Muscle homogenates were prepared, and puromycin-labeled peptides were detected via western blotting.

## Muscle fiber cross-sectional area (CSA)

Ten-micron cryosections of MG and TA muscles were stained with anti-laminin antibody (L9393) and visualized using Alexa Fluor 488-conjugated secondary antibody (A-11008, Thermo Fisher Scientific) [25]. Images were captured using a confocal microscope (FV-3000; Olympus), and CSA was measured using MyoVision software [26]. Sample sizes ranged from 2,000–7,500 fibers per group.

## Maximal torque assessment

Maximal isometric ankle torque was measured 24 h before dissection using a torque dynamometer [27]. Rats were anesthetized with isoflurane (2.0–2.2% in air at 450 mL/min) and placed prone on a platform with the ankle joint positioned at 90°. After shaving the skin, a pair of surface electrodes (Vitrode V, 7.5 × 7.5 mm; Nihon Kohden, Japan) were placed over

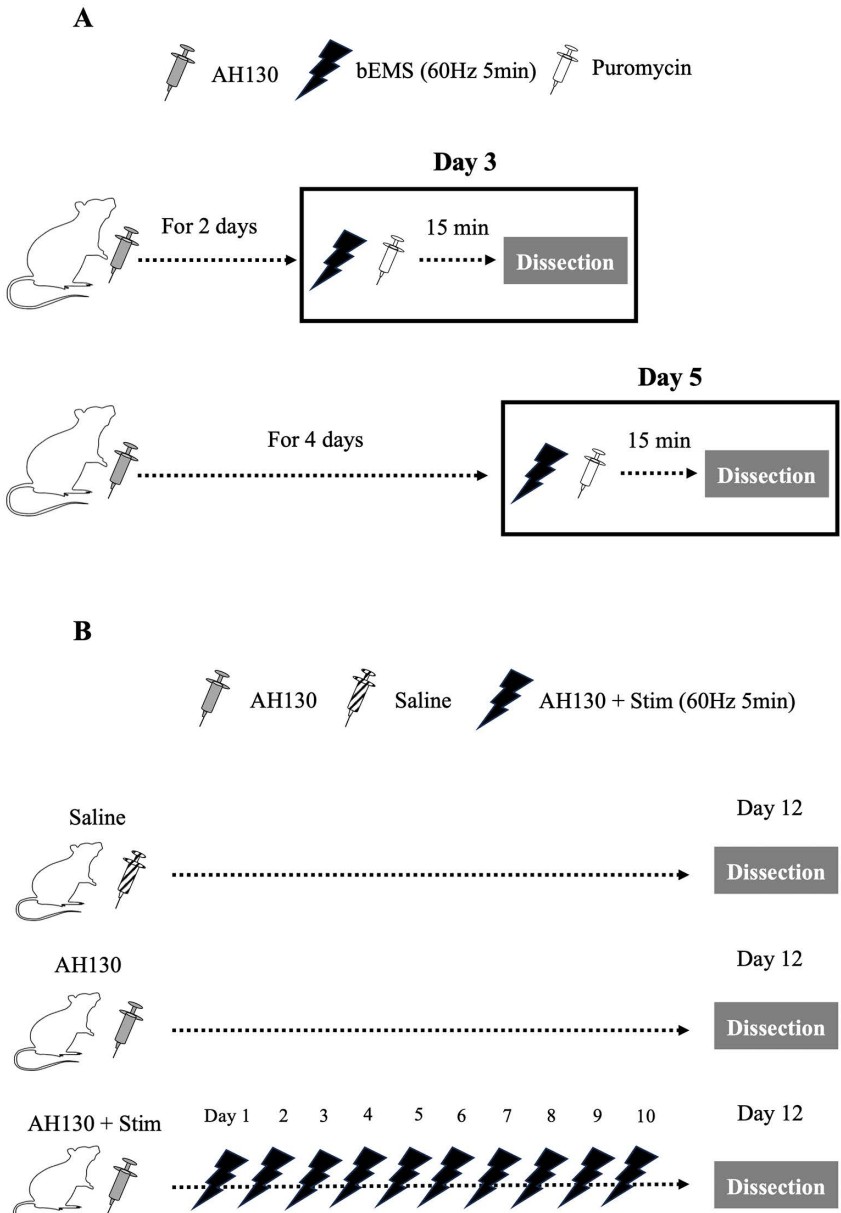

**Fig 2. Experimental design. (A)** Acute response study. Rats with AH130-induced cachexia underwent a single bEMS session. Medial gastrocnemius (MG) and tibialis anterior (TA) muscles were dissected 6 hours post-stimulation to assess mTORC1 signaling and protein synthesis (Control limb [Con], n = 5; Stimulated limb [Stim], n = 5 per time point). **(B)** Chronic response study. Rats were divided into three groups: Saline (n = 7), AH130 (n = 8), and AH130 + Stim (n = 8). The AH130 + Stim group received bEMS for 10 days. Tissues were harvested 48 hours after the final bEMS session.

the gastrocnemius muscle. Maximal isometric torque of the plantarflexor muscles was elicited by percutaneous electrical stimulation of the tibial nerve. Muscle contraction was induced by supramaximal electrical stimulation (100 Hz, 0.4 ms pulse duration, 35 V) delivered from an electrical stimulator with an isolator (Nihon Kohden, Japan). Three trials were performed, and the resulting torque values were averaged for analysis. We measured only plantarflexion torque because the dynamometer setup was designed exclusively for this assessment.

## Statistical analysis

Data are presented as mean ± SEM. Paired Student's t-tests were used for acute comparisons. One-way ANOVA followed by Tukey's post hoc test was used for chronic comparisons of body weight, food intake, CSA, and torque. Average CSA was calculated by Kruskal-Wallis test. Statistical significance was set at $p < 0.05$. Analyses were performed using Graph-Pad Prism (version 8.3.0; La Jolla, CA, USA).

## Results

Acute effects of bEMS on muscle protein synthesis to evaluate the early anabolic effects of bEMS in cancer cachexia, protein synthesis was measured using puromycin incorporation. On day 3 after AH130 cell injection, protein synthesis was significantly higher in the stimulated limbs compared to the contralateral controls in both the medial gastrocnemius (MG) (~1.5-fold, $p < 0.05$, $d = 1.33$; Fig 3A–B) and tibialis anterior (TA) (~2-fold, $p < 0.05$, $d = 1.61$; Fig 4A–B). By day 5, no significant differences were observed between groups in either muscle (MG: $p = 0.23$, $d = 0.64$, TA: $p = 0.42$, $d = 0.40$; Fig 5A–B, 6A–B).

mTORC1 signaling activation following bEMS to assess the involvement of mTORC1 signaling, phosphorylation levels of p70S6K (Thr389) and rpS6 (Ser240/244) were analyzed. A trend toward increased p70S6K phosphorylation was observed in both MG ($p = 0.0725$, $d = 1.08$; Fig 3C) and TA ($p = 0.0714$, $d = 1.09$; Fig 4C) at day 3, though not statistically significant. In contrast, rpS6 phosphorylation was significantly increased in MG at both days 3 ($p < 0.05$, $d = 2.70$) and 5 ($p < 0.05$, $d = 1.59$; Fig 3D, 5D), while no change was observed in TA on either day 3 ($p = 0.10$, $d = 0.97$) and day 5 ($p = 0.95$, $d = 0.03$; Fig 4D, 6D).

Body weight, food intake, and adipose tissue changes after chronic bEMS During the 10-day intervention period, bEMS was applied bilaterally to the lower limbs in the AH130 + Stim group. Neither the AH130 nor AH130 + Stim groups showed increases in body weight over time ($p < 0.05$, $d = 0.41$; Fig 7A). Final body weights were approximately 15–20% lower than saline controls in both groups ($p < 0.05$, $d = 0.50$; Fig 7B–C). Food intake progressively declined in AH130-treated animals (Fig 7D), with significantly reduced cumulative intake ($p < 0.05$, $d = 0.64$; Fig 7E) and inguinal white adipose tissue (iWAT) weight ($p < 0.05$, $d = 0.35$; Fig 7F). Slight attenuation of these effects was observed in the AH130 + Stim group, suggesting a modest protective effect of bEMS on cachexia-induced metabolic changes.

Muscle fiber preservation following chronic bEMS Wet weights of the MG and TA muscles were significantly reduced ($p < 0.05$, MG: $d = 0.46$, TA: $d = 0.31$) in the AH130 and AH130 + Stim groups compared to saline controls (Fig 8A, 8D). Muscle fiber cross-sectional area (CSA) was then assessed to evaluate atrophic changes. In the AH130 + Stim group, CSA of both MG and TA was preserved to a greater extent than in the untreated AH130 group (MG: d = 0.05, TA: 0.06; Fig 8B–C, 8E–F). No significant CSA reduction was observed in the stimulated group compared to saline controls, indicating that bEMS may have partially suppressed muscle fiber atrophy (Fig 8G).

Muscle strength was not improved by bEMS Maximal isometric ankle torque was measured to assess muscle function. Both AH130 and AH130 + Stim groups exhibited reduced torque values compared to saline controls ($p < 0.05$, $d = 0.27$, Fig 9A). When normalized to MG wet weight, no significant differences were detected among groups ($d = 0.07$; Fig 9B), indicating that the 10-day bEMS protocol did not enhance functional strength.

## Discussion

This study investigated the effects of belt-type electrical muscle stimulation (bEMS) on skeletal muscle protein synthesis, fiber morphology, and functional strength in a rat model of cancer cachexia induced by AH130 Yoshida ascites hepatoma cells [20,28,29]. The findings indicate that acute bEMS transiently enhanced muscle protein synthesis, and chronic stimulation partially preserved muscle fiber cross-sectional area (CSA). However, no improvement in muscle strength was observed.

In the acute experiments, on day 3 post-inoculation, protein synthesis was significantly elevated in both the gastrocnemius (MG) and tibialis anterior (TA) muscles following bEMS. This suggests that electrically induced contractions may

**Gastrocnemius (Day 3 after AH130 injection)**

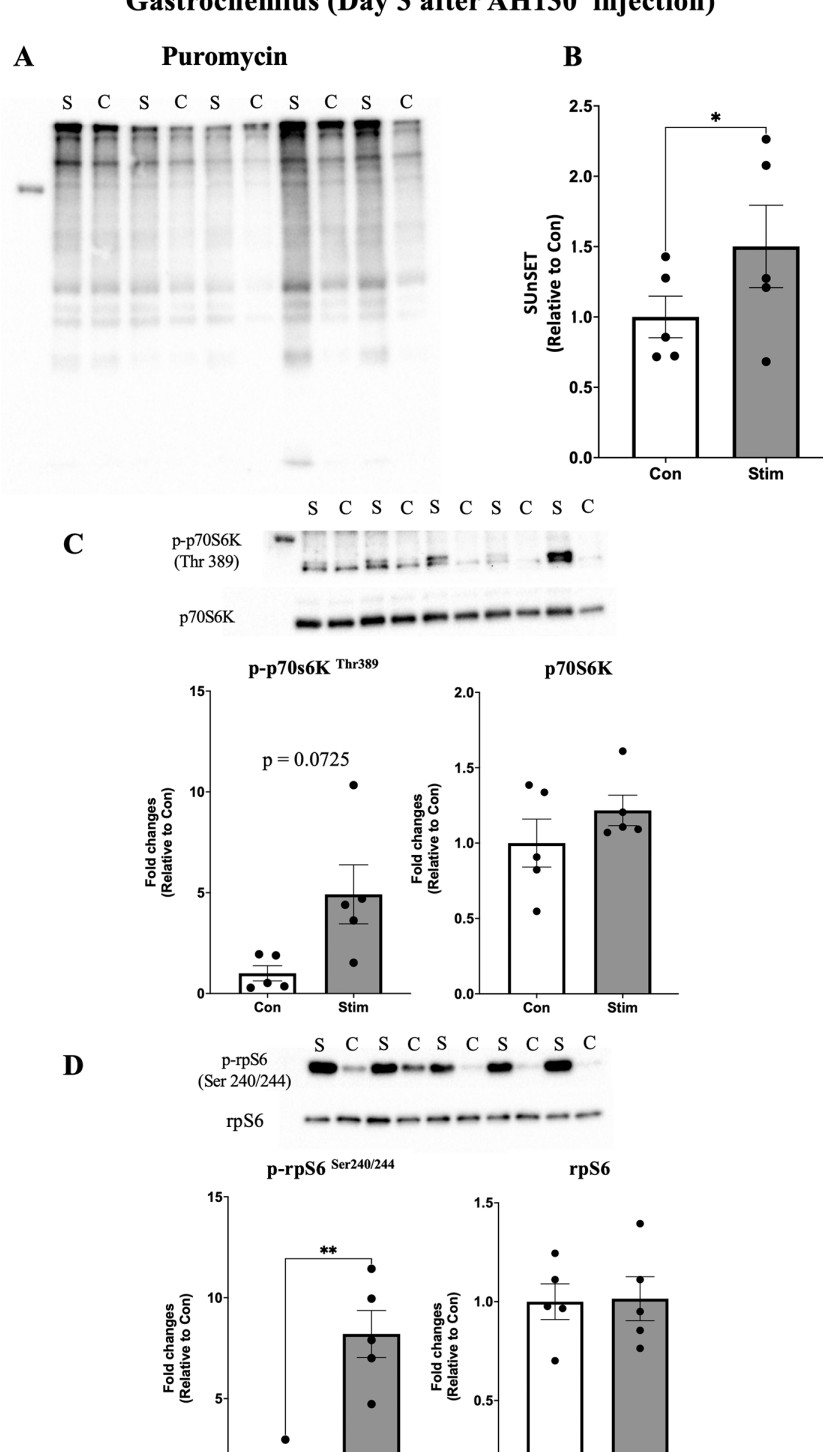

**Fig 3. Acute effects of bEMS on protein synthesis and mTORC1 signaling in the gastrocnemius muscle on day 3 post-AH130 injection. (A)** Representative western blot images. **(B)** Quantification of puromycin-labeled proteins. **(C)** Quantification of phosphorylated p70S6K (Thr389) normalized to total p70S6K. **(D)** Quantification of phosphorylated rpS6 (Ser240/244) normalized to total rpS6. Data are presented as mean±SEM. Comparisons between control (Con) and stimulated (Stim) limbs were made using a paired t-test. *p<0.05 vs. Con.

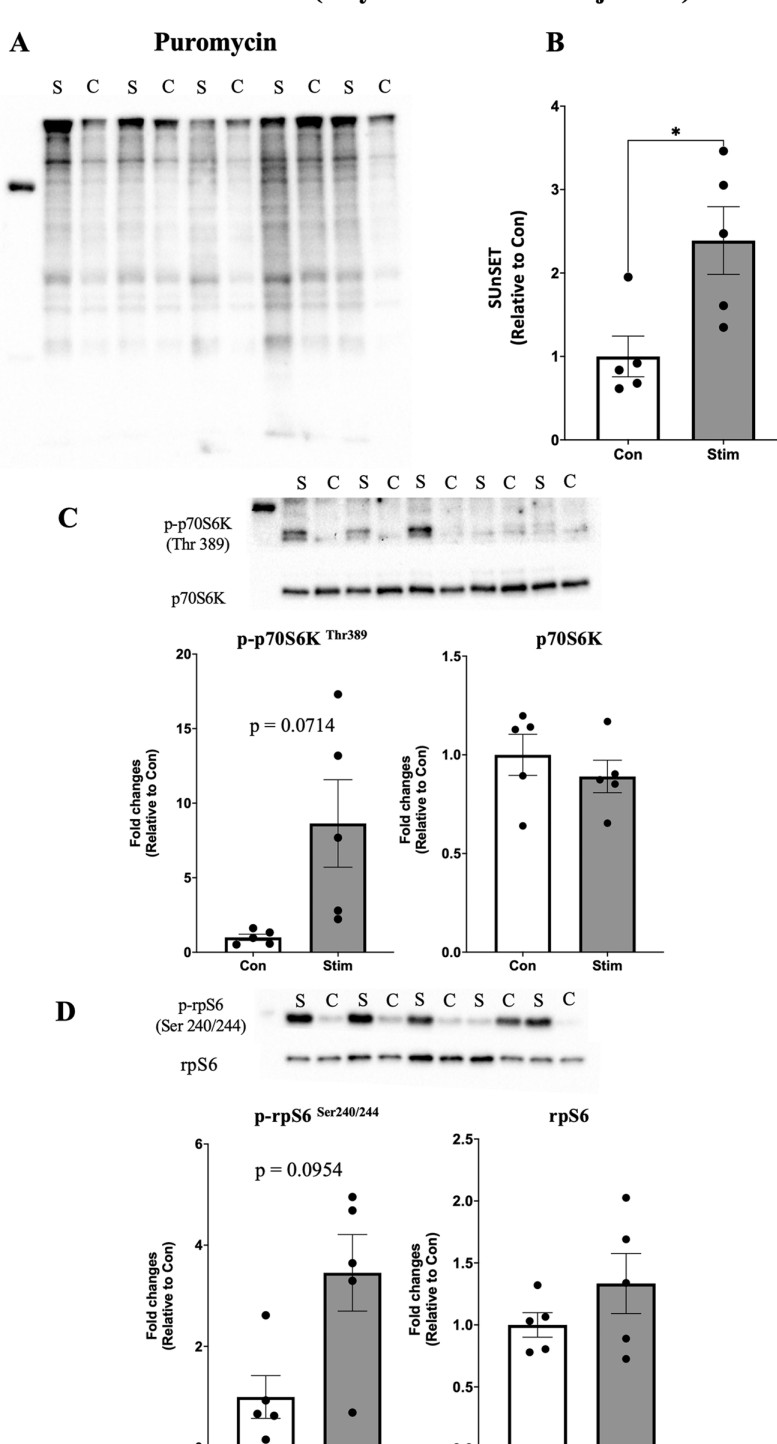

**Tibialis anterior (Day 3 after AH130 injection)**

**Fig 4. Acute effects of bEMS on protein synthesis and mTORC1 signaling in the tibialis anterior muscle on day 3 post-AH130 injection. (A)** Representative western blot images. **(B)** Quantification of puromycin-labeled proteins. **(C)** Quantification of phosphorylated p70S6K (Thr389) normalized to total p70S6K. **(D)** Quantification of phosphorylated rpS6 (Ser240/244) normalized to total rpS6. Data are presented as mean±SEM. Comparisons between control (Con) and stimulated (Stim) limbs were made using a paired t-test. *$p < 0.05$ vs. Con.

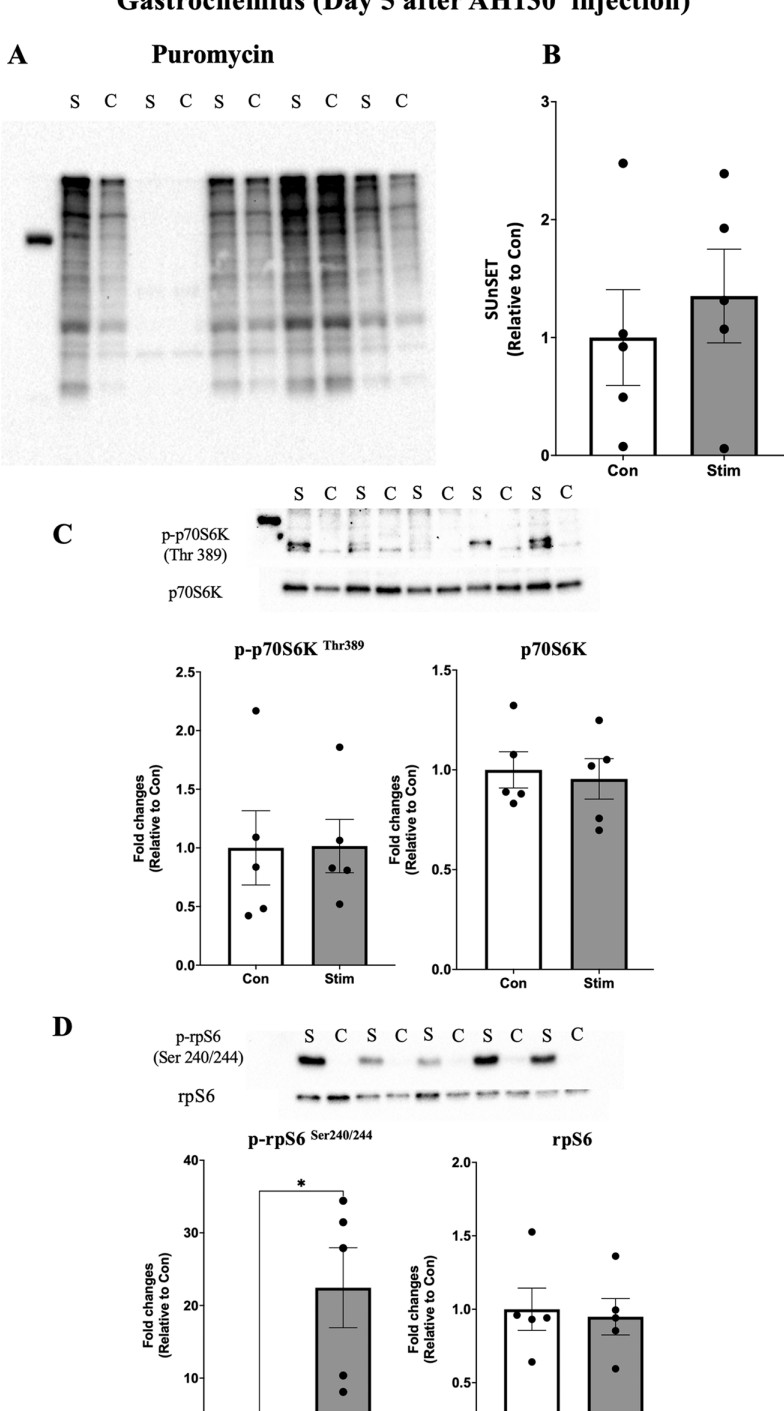

**Gastrocnemius (Day 5 after AH130 injection)**

**Fig 5. Acute effects of bEMS on protein synthesis and mTORC1 signaling in the gastrocnemius muscle on day 5 post-AH130 injection. (A)** Representative western blot images. **(B)** Quantification of puromycin-labeled proteins. **(C)** Quantification of phosphorylated p70S6K (Thr389) normalized to total p70S6K. **(D)** Quantification of phosphorylated rpS6 (Ser240/244) normalized to total rpS6. Data are presented as mean±SEM. Comparisons between control (Con) and stimulated (Stim) limbs were made using a paired t-test. **p<0.01 vs. Con.

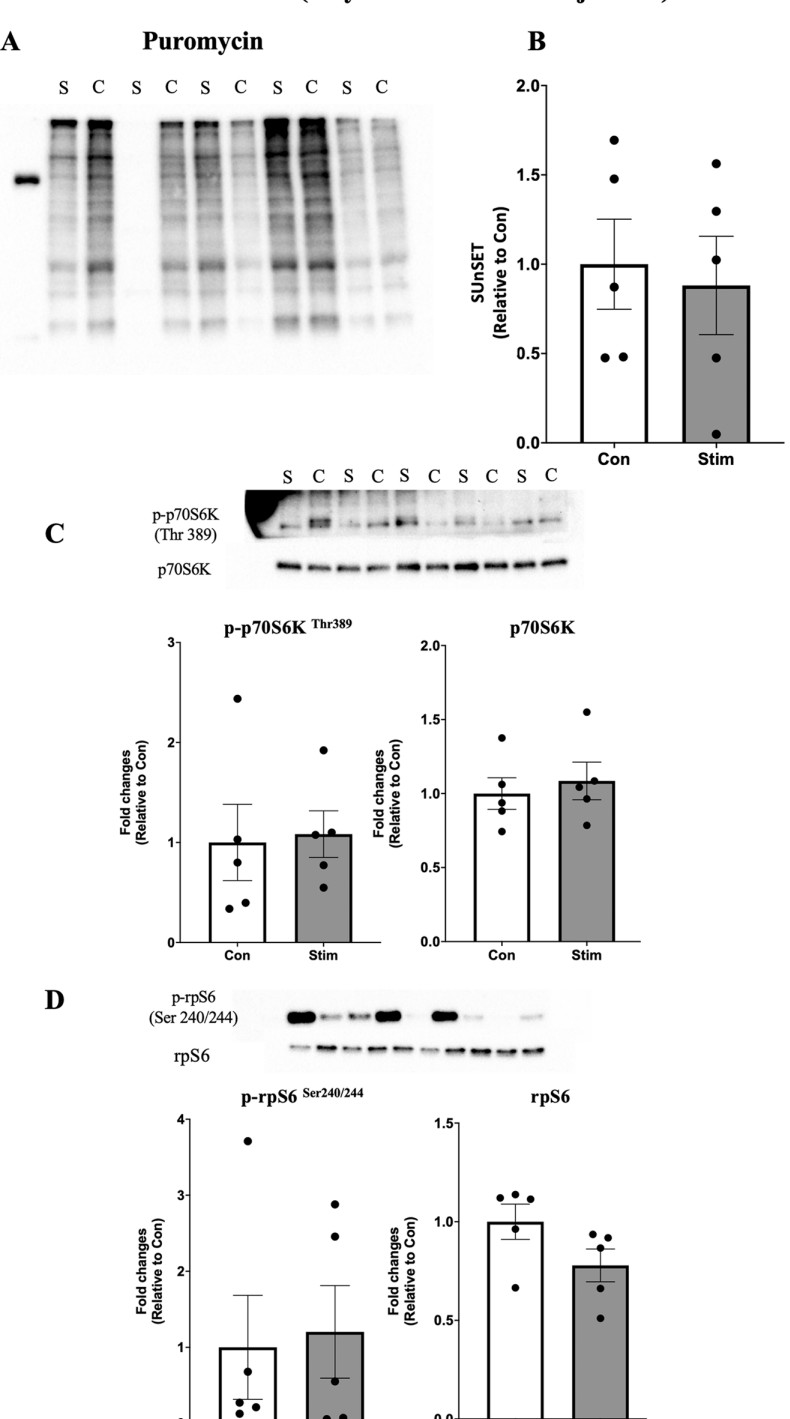

**Tibialis anterior (Day 5 after AH130 injection)**

**Fig 6. Acute effects of bEMS on protein synthesis and mTORC1 signaling in the tibialis anterior muscle on day 5 post-AH130 injection. (A)** Representative western blot images. **(B)** Quantification of puromycin-labeled proteins. **(C)** Quantification of phosphorylated p70S6K (Thr389) normalized to total p70S6K. **(D)** Quantification of phosphorylated rpS6 (Ser240/244) normalized to total rpS6. Data are presented as mean ± SEM. Comparisons between control (Con) and stimulated (Stim) limbs were made using a paired t-test.

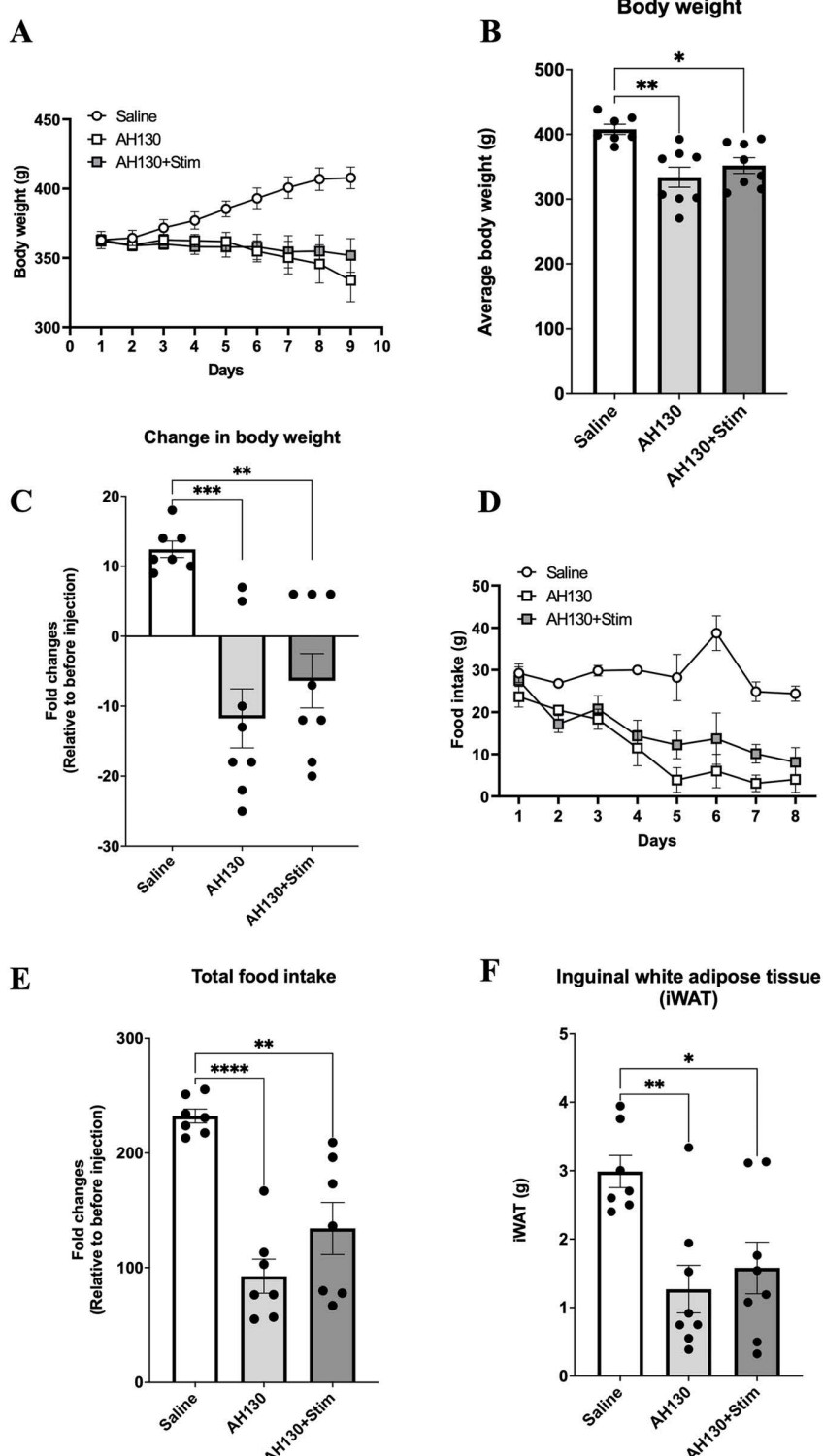

**Fig 7. Effects of chronic bEMS on body weight, food intake, and adipose tissue. (A)** Body weight changes over the 10-day period. **(B)** Final body weights. **(C)** Net change in body weight. **(D)** Daily food intake. **(E)** Total food intake over the experimental period. **(F)** Inguinal white adipose tissue (iWAT) weight. Data are presented as mean±SEM. Differences were assessed by one-way ANOVA with Tukey's post-hoc test. *$p < 0.05$, **$p < 0.01$, ***$p < 0.001$, ****$p < 0.0001$ vs. Saline group.

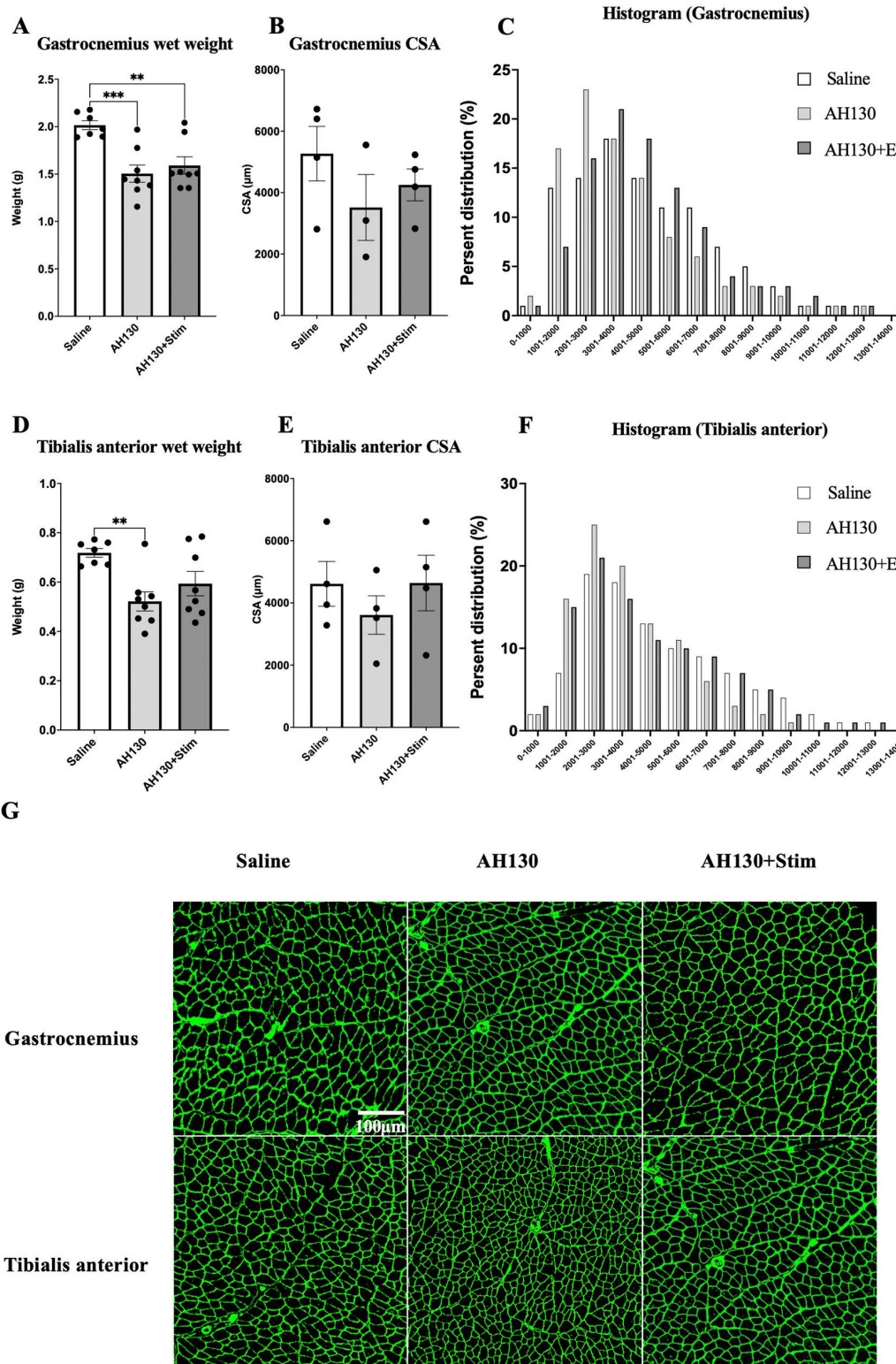

**Fig 8. Effects of chronic bEMS on muscle weight and fiber cross-sectional area (CSA). (A, D)** Wet weights of the MG and TA muscles. **(B, E)** Average muscle fiber CSA. **(C, F)** Histograms showing the distribution of fiber CSA. **(G)** Representative immunofluorescence images of laminin-stained muscle cross-sections. Scale bar applies to all images. Data are presented as mean ± SEM. Differences were assessed by one-way ANOVA with Tukey's post-hoc test. **p < 0.01, ***p < 0.001.

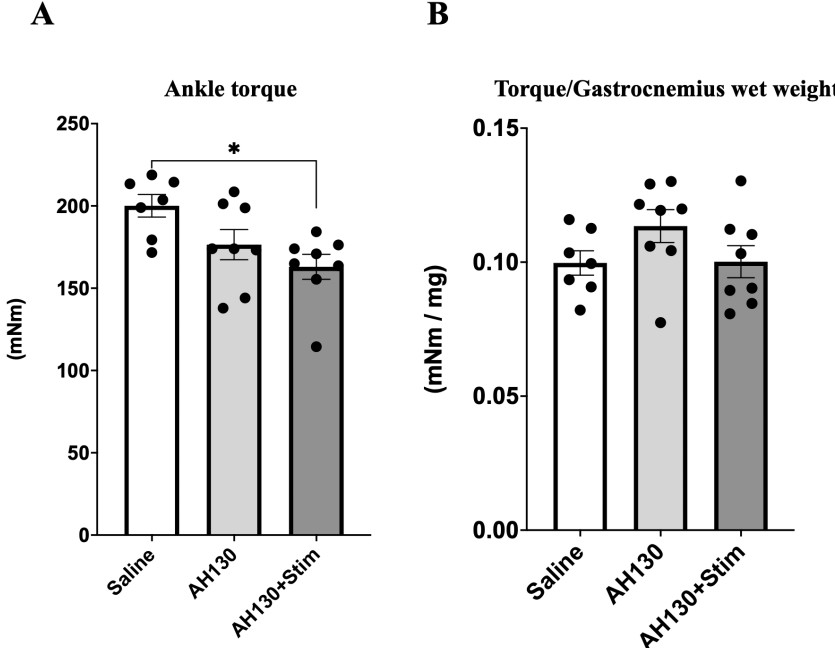

**Fig 9. Effects of chronic bEMS on ankle joint torque. (A)** Maximal isometric torque of the right ankle joint. **(B)** Torque normalized to the combined wet weight of the gastrocnemius muscle. Data are presented as mean±SEM. Differences were assessed by one-way ANOVA with Tukey's post-hoc test. *p<0.05 vs. Saline group.

activate anabolic pathways, including mTORC1 signaling [30,21]. The observed increase in rpS6 phosphorylation supports this interpretation. However, no significant anabolic response was detected by day 5, suggesting that the effect may be short-lived. This attenuation could be related to progressive anabolic resistance associated with cachexia, muscle adaptation to repeated stimulation, or increased systemic inflammation. Cytokines such as TNF-α and IL-6 are known to suppress mTORC1 signaling [31] and may have contributed to the diminished response over time [32].

Chronic bEMS modestly attenuated muscle fiber atrophy in both MG and TA. While muscle CSA was preserved relative to untreated cachectic animals, this effect did not reach statistical significance. Cachexia-induced muscle loss is primarily driven by increased protein degradation via the ubiquitin-proteasome and autophagy-lysosome pathways [11,12].

Although bEMS may modulate these pathways, the current data do not allow definitive conclusions. Despite morphological preservation, muscle strength assessed by ankle plantarflexion torque did not improve with bEMS. This dissociation between preserved muscle fiber size and unimproved function is likely multifactorial. A primary factor is methodological, arising from a key difference with conventional pad-type EMS. While pad-type EMS provides localized stimulation, the bEMS setup concurrently stimulated both agonist (e.g., gastrocnemius) and antagonist (e.g., tibialis anterior) muscles, inducing co-contraction. This inherent to the bEMS setup, likely masked any true functional gains by creating a braking torque [33]. Additionally, the severe inflammatory state of the AH130 model also likely impaired contractile function at the fiber level [34]. Thus, the lack of functional improvement likely reflects a combination of these measurement limitations and underlying deficits in muscle fiber function that were not reversed by bEMS.

## Conclusion

In summary, bEMS transiently increased muscle protein synthesis and partially preserved muscle fiber size in a rat model of cancer cachexia. However, no functional improvement in muscle strength was observed. These findings suggest that

while bEMS may mitigate structural muscle loss, its impact on muscle function is limited. Further studies are needed to clarify its mechanisms and optimize stimulation protocols for potential clinical applications.

## Supporting information

**S1 Fig. Original blots and Ponceau S stains.** (This would be included in the submission as a supplementary file). (PPTX)

## Author contributions

**Conceptualization:** KARINA KOUZAKI, Koichi Nakazato.

**Data curation:** KARINA KOUZAKI, Hiroyuki Uno.

**Formal analysis:** Mako Isemura, Hiroyuki Uno.

**Funding acquisition:** Koichi Nakazato.

**Investigation:** KARINA KOUZAKI, Mako Isemura.

**Methodology:** Mako Isemura, Yuki Tamura, Shunta Tadano, Ryuji Akimoto, Katsu Hosoki.

**Project administration:** KARINA KOUZAKI, Koichi Nakazato.

**Resources:** Hiroyuki Uno, Shunta Tadano, Ryuji Akimoto, Katsu Hosoki.

**Software:** Yuki Tamura, Hiroyuki Uno, Shunta Tadano, Ryuji Akimoto, Katsu Hosoki.

**Supervision:** Yuki Tamura, Koichi Nakazato.

**Validation:** KARINA KOUZAKI, Mako Isemura.

**Visualization:** Hiroyuki Uno.

**Writing – original draft:** KARINA KOUZAKI.

**Writing – review & editing:** Yuki Tamura, Koichi Nakazato.

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
