## [Decision Letter · Decision Letter 0]

3 Sep 2025

Dear Dr. KOUZAKI,

Thank you for submitting your manuscript to PLOS ONE. After careful consideration, we feel that it has merit but does not fully meet PLOS ONE’s publication criteria as it currently stands. Therefore, we invite you to submit a revised version of the manuscript that addresses the points raised during the review process.

Your manuscript has been evaluated by two experts, who have expressed interest in your work. However, they also raised several concerns. We kindly ask you to address each of these points in detail.

We look forward to receiving your revised manuscript.

Kind regards,

Keisuke Hitachi

Academic Editor

PLOS ONE

Journal Requirements:

4. Please ensure that you refer to Figure 1, 2, 6 in your text as, if accepted, production will need this reference to link the reader to the figure.

Reviewers' comments:

Reviewer's Responses to Questions

**Comments to the Author**

1. Is the manuscript technically sound, and do the data support the conclusions?

Reviewer #1: Partly

Reviewer #2: Yes

2. Has the statistical analysis been performed appropriately and rigorously?

Reviewer #1: Yes

Reviewer #2: Yes

3. Have the authors made all data underlying the findings in their manuscript fully available?

Reviewer #1: Yes

Reviewer #2: Yes

4. Is the manuscript presented in an intelligible fashion and written in standard English?

Reviewer #1: Yes

Reviewer #2: Yes

Reviewer #1: General Comments

This study presents the interesting finding that in a rat model of cancer cachexia, belt-type electrical muscle stimulation (bEMS) acutely increases muscle protein synthesis and partially maintains muscle fiber cross-sectional area (CSA), while failing to improve muscle function.

However, several critical questions remain regarding the interpretation of the results and the novelty of the study. The most important issue is that it is unclear whether the observed outcome—the discrepancy between maintained muscle mass and a lack of improvement in strength—is a phenomenon specific to the bEMS method or a general response inherent to the cancer cachexia model itself. As the authors note, a key feature of bEMS is its ability to stimulate a wide area. A more in-depth discussion is needed to connect this characteristic to the results.

If the objective of this study is to investigate the unique properties of bEMS, a direct comparison with a conventional pad-type EMS that involves joint movement, for example, would provide a clear answer to this question. If a comparative experiment is not feasible, it is necessary to provide specific details on the characteristics of bEMS-induced muscle activity, such as its range and its effects on both agonist and antagonist muscles, and to discuss how these factors influenced the results.

Specific points are detailed below.

Major Points

Introduction

The structure of the Introduction is redundant. Similar descriptions regarding the pathophysiology of cachexia (e.g., prevalence, protein metabolism imbalance) are repeated in the first and second halves of the section. It is recommended to consolidate this content and revise it for a more concise and logical flow.

For example:

In cancer patients, cachexia affects approximately 50–80% of individuals [11], compared to 5–15% in other chronic diseases [7]. The syndrome is driven by an imbalance between protein synthesis and degradation, with increased proteolysis [9, 22] and impaired anabolic signaling contributing to progressive muscle atrophy [4].

Electrical muscle stimulation (EMS) is a non-pharmacological intervention capable of eliciting muscle contractions in the absence of voluntary effort.

Among these features, skeletal muscle wasting is particularly problematic due to its association with impaired physical function and increased mortality. In cancer patients, cachexia has been reported in 50–80% of cases [11], significantly exceeding its prevalence in other chronic conditions. The underlying mechanisms involve increased protein degradation and reduced muscle protein synthesis, resulting in net muscle loss [4, 9, 22].

Electrical muscle stimulation (EMS) is a passive intervention that induces muscle contraction without voluntary movement.

Methods

The description of the experimental protocol is insufficient, making it difficult for readers to accurately understand and replicate the experiment. Please add detailed information on the following points in particular:

Electrical Stimulation Protocol:

Quantification of Exercise Load: The term "sustained tetanic contraction" is used, but it should be specified what percentage of maximal muscle strength this load corresponds to. Alternatively, the objective criteria used to determine the stimulation intensity should be clearly stated.

Presence/Absence of Joint Movement: Was the ankle joint immobilized during stimulation, or was joint movement (plantarflexion/dorsiflexion) involved? It must be clearly stated whether the contractions were isometric or isotonic.

Effect of Simultaneous Antagonist Stimulation: A key characteristic of bEMS is the simultaneous stimulation of antagonist muscles. The theoretical impact of this on force production should be mentioned in the Methods section.

Maximal Torque Measurement:

Stimulation Method: What type of electrical stimulation was used to elicit maximal torque? Please be specific, for instance, whether it was percutaneous nerve stimulation.

Stimulation Parameters: Could you provide the optimized stimulation parameters (e.g., frequency) used to elicit maximal tension, including whether frequency-dependence was considered?

Measured Parameters: In relation to Figure 9, was only plantarflexion torque measured? The reason for not measuring the corresponding antagonist (dorsiflexion) torque should also be clarified.

Discussion

In the Discussion, you mention that the simultaneous contraction (co-contraction) of agonist and antagonist muscles may have masked functional gains. However, the explanation of the physiological mechanism is insufficient. Please provide a specific description of the mechanism by which co-contraction reduces net joint torque and include appropriate citations to support this argument.

Reviewer #2: The purpose of this study was to examine the impacts of belt-type electrical muscle stimulation (bEMS) on protein synthesis, muscle fiber cross-sectional area, and muscle function in rat model of cancer cachexia. The authors demonstrated that bEMS can be a supportive strategy to prevent muscle atrophy in cachexia. Basically, the topic of this study is interesting. However, I have several comments and concerns that will strengthen the manuscript. Please see below.

In Introduction, I understand that the authors conducted both an acute and a chronic study. However, the rationale for this design remains unclear. Particularly, the duration of the two experiments does not appear to be different, which makes the distinction between "acute" and "chronic" ambiguous. A more detailed explanation of the purpose of each study and the justification for using two distinct protocols would be beneficial to readers.

Sentences after line 63 may be excluded. I guess these are repetitions of the previous parts.

It is suggested that the authors report the characteristics of AH130 cell-line in the Introduction to provide a more comprehensive background on this model.

In Material and Methods, in my understanding, the authors simulated single leg in the acute study. Conversely, EMS was applied bilaterally in the chronic study. Please clarify the reason for this difference in the EMS application protocol between the two studies.

Please provide a rationale for the selection of the medial gastrocnemius (MG) and tibialis anterior (TA) muscles for measurement.

In Results, the authors suggest that bEMS has a protective effect on some of the measured variables. However, some of the results are not statistically significant. To support this notion, it is recommended that the authors report the effect size, which may provide an additional interpretation of the findings.

In Discussion, lines 227-229: These points are important for interpreting the present results. Please provide more details about the characteristics of the cachectic model and offer a more elaborate discussion on these findings.

**Do you want your identity to be public for this peer review?** For information about this choice, including consent withdrawal, please see our Privacy Policy

Reviewer #1: No

Reviewer #2: No

---

## [Author Response · Author response to Decision Letter 1]

15 Oct 2025

October/15/2025

PLOS ONE Editorial Office

PONE-D-25-40946

Title: Belt-type electrical muscle stimulation preserves muscle fiber size but does not improve muscle function in a rat model of cancer cachexia

Corresponding Author: Karina Kouzaki

Authors: Karina Kouzaki, Mako Isemura, Yuki Tamura, Hiroyuki Uno, Shunta Tadano, Ryuji Akimoto, Katsu Hosoki, and Koichi Nakazato

Dear Editor in PLOS ONE:

Thank you for your letter dated October, 15th 2025 that was received along with the reviewers' comments. We would like to express our sincere apologies for the delay in the resubmission of our revised manuscript. We are deeply grateful for your patience and understanding during this extended revision period. We have used this time to undertake a comprehensive re-evaluation of the manuscript's structure and have made significant revisions to improve its clarity and organization. We were pleased to learn that our manuscript was rated as potentially acceptable for publication in the PLOS ONE, subject to adequate revision and response to the comments and concerns raised by the reviewers. We have diligently addressed all of the comments and suggestions made by Reviewers, and a detailed, point-by-point response to their feedback is provided.

We sincerely appreciate the reviewers' insightful comments and suggestions. We would like to thank the editor and the reviewers for their valuable comments and suggestions, and we hope that the revised version is acceptable for publication in PLOS ONE.

Sincerely,

Karina Kouzaki

Nippon Sport Science University, 7-1-1, Fukasawa, Setagaya-ku, Tokyo,158-8508, Japan

E-mail address: kouzaki@nittai.ac.jp

Reply to Reviewer # 1

Thank you very much for reviewing our manuscript and providing valuable and constructive comments that have helped us to improve the manuscript. We have responded to all of the points that were raised by you, the other reviewer, and the editor, and revised the manuscript accordingly.

We hope that the revisions will satisfy your standards.

We greatly appreciate your further review of the revised manuscript.

Our responses to your comments and suggestions (shown in Blue text) are as follows.

Dear Reviewer 1

General Comments

This study presents the interesting finding that in a rat model of cancer cachexia, belt-type electrical muscle stimulation (bEMS) acutely increases muscle protein synthesis and partially maintains muscle fiber cross-sectional area (CSA), while failing to improve muscle function.

However, several critical questions remain regarding the interpretation of the results and the novelty of the study. The most important issue is that it is unclear whether the observed outcome—the discrepancy between maintained muscle mass and a lack of improvement in strength—is a phenomenon specific to the bEMS method or a general response inherent to the cancer cachexia model itself. As the authors note, a key feature of bEMS is its ability to stimulate a wide area. A more in-depth discussion is needed to connect this characteristic to the results.

Thank you for your constructive review. As you suggested, we have revised the Discussion section to carefully consider whether the observed discrepancy between maintained muscle mass and unimproved strength is an inherent characteristic of bEMS, or a phenomenon unique to cancer cachexia. Additionally, we found that some in-text citation numbers did not correspond with the reference list. We have therefore renumbered all citations throughout the manuscript to correct this.

If the objective of this study is to investigate the unique properties of bEMS, a direct comparison with a conventional pad-type EMS that involves joint movement, for example, would provide a clear answer to this question. If a comparative experiment is not feasible, it is necessary to provide specific details on the characteristics of bEMS-induced muscle activity, such as its range and its effects on both agonist and antagonist muscles, and to discuss how these factors influenced the results.

Specific points are detailed below.

Unfortunately, a direct comparison with pad-type EMS was not feasible within the scope of this study. Instead, we have provided a discussion on the specific characteristics of muscle activity from bEMS and how these factors likely influenced the results. Please see pages 11-13, Discussion section

Major Points

Introduction

The structure of the Introduction is redundant. Similar descriptions regarding the pathophysiology of cachexia (e.g., prevalence, protein metabolism imbalance) are repeated in the first and second halves of the section. It is recommended to consolidate this content and revise it for a more concise and logical flow.

For example:

In cancer patients, cachexia affects approximately 50–80% of individuals [11], compared to 5–15% in other chronic diseases [7]. The syndrome is driven by an imbalance between protein synthesis and degradation, with increased proteolysis [9, 22] and impaired anabolic signaling contributing to progressive muscle atrophy [4].

Electrical muscle stimulation (EMS) is a non-pharmacological intervention capable of eliciting muscle contractions in the absence of voluntary effort.

Among these features, skeletal muscle wasting is particularly problematic due to its association with impaired physical function and increased mortality. In cancer patients, cachexia has been reported in 50–80% of cases [11], significantly exceeding its prevalence in other chronic conditions. The underlying mechanisms involve increased protein degradation and reduced muscle protein synthesis, resulting in net muscle loss [4, 9, 22].

Electrical muscle stimulation (EMS) is a passive intervention that induces muscle contraction without voluntary movement.

We agree that the introduction was redundant. We have substantially shortened the introduction in the revised manuscript.

Please see pages 2-3, Introduction section

Methods

The description of the experimental protocol is insufficient, making it difficult for readers to accurately understand and replicate the experiment.

Please add detailed information on the following points in particular:

Thank you for this valuable feedback. We have carefully revised the methods section according to your suggestions.

Electrical Stimulation Protocol:

Quantification of Exercise Load: The term "sustained tetanic contraction" is used, but it should be specified what percentage of maximal muscle strength this load corresponds to. Alternatively, the objective criteria used to determine the stimulation intensity should be clearly stated.

Presence/Absence of Joint Movement: Was the ankle joint immobilized during stimulation, or was joint movement (plantarflexion/dorsiflexion) involved? It must be clearly stated whether the contractions were isometric or isotonic.

We have added a description of the electrical stimulation protocol. Please see pages 5-6, lines 103-105.

Effect of Simultaneous Antagonist Stimulation: A key characteristic of bEMS is the simultaneous stimulation of antagonist muscles. The theoretical impact of this on force production should be mentioned in the Methods section.

We fully agree with your comment. In accordance with your point, we have added about the following sentence regarding the effect of antagonist muscle stimulation: “A key characteristic of This bEMS setup is the simultaneous stimulation of antagonist muscles (e.g., tibialis anterior) along with agonist muscles (e.g., gastrocnemius), which may result in co-contraction and theoretically reduce the net joint torque.” Please see pages 6, lines 107-109

Maximal Torque Measurement:

Stimulation Method: What type of electrical stimulation was used to elicit maximal torque? Please be specific, for instance, whether it was percutaneous nerve stimulation.

Stimulation Parameters: Could you provide the optimized stimulation parameters (e.g., frequency) used to elicit maximal tension, including whether frequency-dependence was considered?

Thank you for these insightful questions, which have helped us improve the clarity of our manuscript. We have revised the Methods section to include the specific details you requested regarding the stimulation method and parameters.

Specifically:

Stimulation Method: We have now explicitly stated that maximal torque was elicited via percutaneous electrical stimulation of the tibial nerve.

Stimulation Parameters: We have clarified that the reported parameters (100 Hz, 0.4 ms pulse duration, 35 V) were those optimized to produce maximal tetanic tension.

We believe the revised text, provided below, now fully addresses your questions.

“Maximal isometric ankle torque was measured 24 h before dissection using a torque dynamometer [35]. Rats were anesthetized with isoflurane (2.0-2.2%) and placed prone on a platform with the ankle joint positioned at 90°. After shaving the skin, a pair of surface electrodes (Vitrode V; Nihon Kohden, Japan) were placed over the gastrocnemius muscle. Maximal isometric torque of the plantarflexor muscles was elicited by percutaneous electrical stimulation of the tibial nerve. Muscle contraction was induced by supramaximal electrical stimulation (100 Hz, 0.4 ms pulse duration, 35 V) delivered from an electrical stimulator with an isolator (Nihon Kohden, Japan). Three trials were performed, and the resulting torque values were averaged for analysis.”

Please see pages 8, lines 149-159

Measured Parameters: In relation to Figure 9, was only plantarflexion torque measured? The reason for not measuring the corresponding antagonist (dorsiflexion) torque should also be clarified.

Thank you for this question for clarification.

Yes, we measured only the plantarflexion torque. The reason for this is a technical limitation of our equipment; the torque dynamometer used in this study is specifically designed for measuring plantarflexion and is not equipped to measure the corresponding dorsiflexion torque.

We have now clarified this point in the Methods section. Please see pages 8-9, lines 158-159

Discussion

In the Discussion, you mention that the simultaneous contraction (co-contraction) of agonist and antagonist muscles may have masked functional gains. However, the explanation of the physiological mechanism is insufficient. Please provide a specific description of the mechanism by which co-contraction reduces net joint torque and include appropriate citations to support this argument.

Thank you for this excellent suggestion. We have revised the Discussion to provide a specific description of the mechanism by which co-contraction reduces net joint torque and have added appropriate citations as requested. Please see pages 13, lines 232-242

Reply to Reviewer # 2

Thank you very much for reviewing our manuscript and providing valuable and constructive comments that have helped us to improve the manuscript. We have responded to all of the points that were raised by you, the other reviewer, and the editor, and revised the manuscript accordingly.

We hope that the revisions will satisfy your standards.

We greatly appreciate your further review of the revised manuscript.

Our responses to your comments and suggestions (shown in Blue text) are as follows.

The purpose of this study was to examine the impacts of belt-type electrical muscle stimulation (bEMS) on protein synthesis, muscle fiber cross-sectional area, and muscle function in rat model of cancer cachexia. The authors demonstrated that bEMS can be a supportive strategy to prevent muscle atrophy in cachexia. Basically, the topic of this study is interesting. However, I have several comments and concerns that will strengthen the manuscript. Please see below.

We thank the reviewer for their positive evaluation and constructive comments. We were pleased to hear that they found the topic interesting. We have addressed each of the concerns raised and believe the manuscript has been significantly strengthened as a result. Additionally, we found that some in-text citation numbers did not correspond with the reference list. We have therefore renumbered all citations throughout the manuscript to correct this. Please find our detailed responses below.

In Introduction, I understand that the authors conducted both an acute and a chronic study. However, the rationale for this design remains unclear. Particularly, the duration of the two experiments does not appear to be different, which makes the distinction between "acute" and "chronic" ambiguous. A more detailed explanation of the purpose of each study and the justification for using two distinct protocols would be beneficial to readers.

Thank you for this insightful comment. We agree that the rationale for our dual-protocol design was not sufficiently explained in the original manuscript, leading to ambiguity between the "acute" and "chronic" phases. We have revised the Introduction to clarify this.

The acute protocol was designed to provide mechanistic insight. Its purpose was to determine if a single session of bEMS could elicit an immediate anabolic response, specifically by measuring muscle protein synthesis, which is a transient event peaking hours after stimulation.

The chronic protocol was designed to assess the long-term adaptive outcomes. Its purpose was to evaluate whether the cumulative effect of repeated sessions over 10 days would translate into tangible structural (muscle fiber CSA) and functional (torque) changes.

Thus, "acute" refers to the immediate response to a single stimulus, while "chronic" refers to the adaptive result of multiple stimuli over time. We believe this distinction is now clearly articulated in the revised Introduction.

Please see pages 3, lines 58-65

Sentences after line 63 may be excluded. I guess these are repetitions of the previous parts.

We agree with your comment, which was also noted by Reviewer 1, that the introduction was redundant. Accordingly, we have revised the entire introduction to be shorter and more focused. Pleas see pages 2-3, Introduction section

It is suggested that the authors report the characteristics of AH130 cell-line in the Introduction to provide a more comprehensive background on this model.

Thank you for this valuable suggestion. We agree that providing a brief background on the AH130 cell line model is important for the reader's comprehension.

As requested, we have now added a sentence to the Introduction describing the characteristics of this well-established cachexia model. Please see pages 3, lines 56-58

In Material and Methods, in my understanding, the authors simulated single leg in the acute study. Conversely, EMS was applied bilaterally in the chronic study. Please clarify the reason for this difference in the EMS application protocol between the two studies.

We thank the reviewer for this helpful suggestion. We have revised the Methods section to clarify the distinct rationale for each protocol:

Acute: To utilize the contralateral limb as an internal control for mechanistic analysis.

Chronic: To assess a potential therapeutic effect while avoiding the confounding variable of muscular imbalance.

Please see pages 6, lines 113-115, 119-121

Please provide a rationale for the selection of the medial gastrocnemius (MG) and tibialis anterior (TA) muscles for measurement.

We selected the medial gastrocnemius (MG) and tibialis anterior (TA) for specific functional and practical reasons.

First, they represent the primary functional antagonist pair for the ankle joint; the MG is a key plantarflexor, and the TA is the primary dorsiflexor. Analyzing both provides a comprehensive assessment of the joint's main functions. Second, from a technical standpoint, their size and distinct anatomical locations in the rat facilitate reliable dissection and provide sufficient tissue for our various analyses.

To make this rationale clear, we have now added a brief explanation to the Methods section. Please see pages 6, lines 107-109

In Results, the authors suggest that bEMS has a protective effect on some of the measured variables. However, some of the results are not statistically significant. To support this notion, it is recommended that the authors report the effect size, which may provide an additional interpretation of the findings.

Thank you

---

## [Decision Letter · Decision Letter 1]

27 Oct 2025

Belt-type electrical muscle stimulation preserves muscle fiber size but does not improve muscle function in a rat model of cancer cachexia

PONE-D-25-40946R1

Dear Dr. KOUZAKI,

We’re pleased to inform you that your manuscript has been judged scientifically suitable for publication and will be formally accepted for publication once it meets all outstanding technical requirements.

Kind regards,

Keisuke Hitachi

Academic Editor

PLOS ONE

Additional Editor Comments (optional):

Reviewers' comments:

Reviewer's Responses to Questions

**Comments to the Author**

Reviewer #1: All comments have been addressed

Reviewer #2: All comments have been addressed

2. Is the manuscript technically sound, and do the data support the conclusions?

Reviewer #1: Yes

Reviewer #2: Yes

3. Has the statistical analysis been performed appropriately and rigorously?

Reviewer #1: Yes

Reviewer #2: Yes

4. Have the authors made all data underlying the findings in their manuscript fully available?

Reviewer #1: Yes

Reviewer #2: Yes

5. Is the manuscript presented in an intelligible fashion and written in standard English?

Reviewer #1: Yes

Reviewer #2: Yes

Reviewer #1: I have reviewed the revised manuscript and confirmed that the points raised in the previous review have been adequately addressed. I have no further comments.

Reviewer #2: (No Response)

**Do you want your identity to be public for this peer review?** For information about this choice, including consent withdrawal, please see our Privacy Policy

Reviewer #1: No

Reviewer #2: No

---

## [Editor Report · Acceptance letter]

PONE-D-25-40946R1

PLOS ONE

Dear Dr. KOUZAKI,

I'm pleased to inform you that your manuscript has been deemed suitable for publication in PLOS ONE. Congratulations! Your manuscript is now being handed over to our production team.

Kind regards,

on behalf of

Dr. Keisuke Hitachi

Academic Editor

PLOS ONE